# TREND: Unsupervised 3D Representation Learning via Temporal Forecasting for LiDAR Perception

**Runjian Chen**[1]    **Hyoungseob Park**[2]    **Bo Zhang**[3]    **Wenqi Shao**[3]
**Ping Luo**[1,4*]    **Alex Wong**[2*]

[1]The University of Hong Kong    [2]Yale University
[3]Shanghai AI Laboratory    [4]HKU Shanghai Intelligent Computing Research Center
{rjchen, pluo}@cs.hku.hk    alex.wong@yale.edu

## Abstract

Labeling LiDAR point clouds is notoriously time-and-energy-consuming, which spurs recent unsupervised 3D representation learning methods to alleviate the labeling burden in LiDAR perception via pretrained weights. Existing work focus on either masked auto encoding or contrastive learning on LiDAR point clouds, which neglects the temporal LiDAR sequence that naturally accounts for object motion (and their semantics). Instead, we propose TREND, short for **T**emporal **RE**ndering with **N**eural fiel**D**, to learn 3D representation via forecasting the future observation in an unsupervised manner. TREND integrates forecasting for 3D pre-training through a Recurrent Embedding scheme to generate 3D embeddings across time and a Temporal LiDAR Neural Field specifically designed for LiDAR modality to represent the 3D scene, with which we compute the loss using differentiable rendering. We evaluate TREND on 3D object detection and LiDAR semantic segmentation tasks on popular datasets, including Once, Waymo, NuScenes, and SemanticKITTI. TREND generally improves from-scratch models across datasets and tasks and brings gains of 1.77% mAP on Once and 2.11% mAP on NuScenes, which are up to $400\%$ more improvement compared to previous SOTA unsupervised 3D pre-training methods. Codes and models will be available here.

## 1 Introduction

Light-Detection-And-Ranging (LiDAR) is widely used in autonomous driving. By emitting laser rays into the environment, it provides accurate measurements of the distance along each ray with time-of-flight principle. There has been strong research interest on LiDAR-based perception like 3D object detection [1, 2, 3, 4, 5, 6, 7] and semantic segmentation [8, 9]. However, labeling for LiDAR point clouds is notoriously time-and-energy-consuming. According to [10], it costs an expert labeler at least 10 minutes to label one frame of LiDAR point cloud at a coarse-level and more at finer granularity. Assuming sensor frequency at $20Hz$, it could cost more than 1000 days of a human expert to annotate a one-hour LiDAR sequence. To alleviate the labeling burden, unsupervised 3D representation learning [11, 12, 13, 14, 15, 16, 17, 18, 19, 20, 21] pre-trains 3D backbone and fine-tune on downstream tasks for performance improvement with the same number of labels.

Previous literature on unsupervised 3D representation learning for LiDAR perception can be divided into two streams, as shown in Figure 1 (a) and (b). (a) Masked-autoencoder-based methods [17, 18, 19, 20, 21] randomly mask LiDAR point clouds and the pre-training entails reconstructing the masked areas. (b) Contrastive-based methods [15, 16] construct two views from one frame (or adjacent frames) of LiDAR point cloud and maximize the similarity among positive pairs while minimizing

---

*Corresponding authors.

39th Conference on Neural Information Processing Systems (NeurIPS 2025).

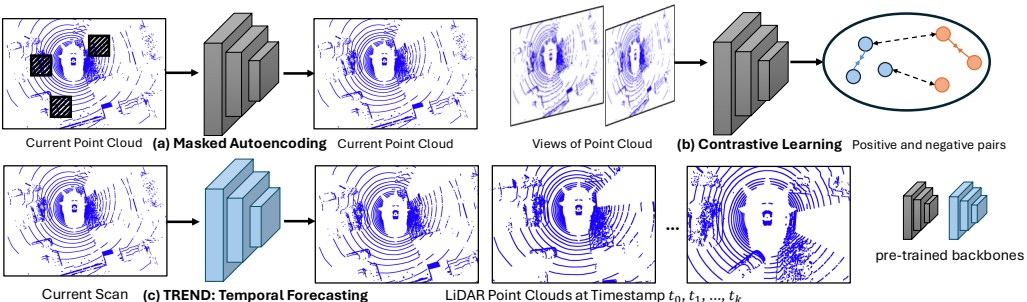

Figure 1: Different schemes for unsupervised 3D representation learning. (a) Masked Autoencoding applies random masking and pre-train by reconstruction. (b) Contrastive methods build views of point cloud and pre-train by pulling together positive pairs and pushing away negative pairs. (c) TREND explores object motion and semantic information via temporal forecasting in LiDAR sequence.

the similarity of negative pairs. Both approaches assume a predefined set of nuisance variability. Nuisance variability refers to variables inherent in the input that should be non-consequential to the outcome, but nonetheless may impact the output. An example of this is orientation: the same object appearing in different orientations can cause the outcome to differ. To obtain the same outcome, one needs to be invariant. In (a), the set of nuisance variability is occlusions, which naturally is induced by motion; in (b) it is the handcrafted set of transformations on LiDAR scans used in contrastive learning. While the procedures are unsupervised, they implicitly select the set of invariants, which benefits the downstream tasks. Unlike them, we subscribe to allowing the data to determine nuisances by simply observing and predicting scene dynamics. This leads to a novel unsupervised 3D representation learning approach based on forecasting LiDAR point clouds (Figure 1 (c)). Naturally, points belonging to the same object instance, within a point cloud, tend to move together. By observing current point cloud and predicting future observation, our pre-training scheme implicitly encodes semantics and biases of object interactions over time.

However, leveraging forecasting as unsupervised 3D representation is nontrivial as scene dynamics are often complex and nonlinear. There are two main challenges: 1) How to generate 3D embeddings at different timestamps with current LiDAR scan? 2) How to represent the 3D scene with embeddings and optimize the network via forecasting?

For 1), there exists tangential work in occupancy prediction field [22, 23, 24] that generates 3D features at different timestamps via directly using 3D/2D convolution [22, 23] or a diffusion decoder with frozen 3D encoder [24]. However, actions of the ego-vehicle is not taken into account in [22, 23], which is important for future observation forecasting as the ego actions reflect the interaction between ego-vehicle and other traffic participants. For example, if the ego-vehicle is running at a high speed, pedestrians might stop to avoid accidents. If the ego-vehicle stops at the crossing, pedestrians might start to walk across the road. For 2), applying neural field, as in existing work [19, 20, 21], to represent the 3D scene at different timestamps yields little to no improvement. The first reason is that the network needs to learn to understand the concept of "time" with the 3D convolution, which could be very difficult. The second one is that the neural fields in [19, 20, 21, 25, 26] are designed for camera modality, which neglects important characteristic in LiDAR modality like intensity.

We address these challenges by proposing TREND, short for **T**emporal **RE**ndering with **N**eural fiel**D**, for unsupervised 3D pre-training. For 1), we propose a Recurrent Embedding scheme, which generates 3D embeddings along time axis with sinusoidal encoding of the ego actions followed by a shallow 3D convolution. This enables us to model ego actions over time, which also assists in forecasting future observations. For 2), we propose a Temporal LiDAR Neural Field that explicit takes timestamps as inputs and integrates LiDAR geometry (surface points) as well as intensity to reconstruct and forecast LiDAR point clouds for optimizing the backbones. While this takes inspiration from existing work in neural field decoders [19, 20, 21, 25, 26], it is distinct from them in that our design enables forecasting and also modeling of LiDAR characteristics, such as intensity.

We demonstrate TREND on four benchmark datasets (Once [27], NuScenes [28], Waymo [29], and Semantic Kitti [30]) for the downstream 3D object detection and LiDAR semantic segmentation tasks,

where TREND achieves up to $400\%$ more improvement compared to previous SOTA pre-training method for Once (1.77% mAP) and improves by $90\%$ on NuScenes (2.11% mAP).

## 2 Related Work

**Pre-training for Point Cloud.** Since annotating 3D point clouds requires significant effort and time, there has been great interest on improving label efficiency for point cloud perception via 3D pre-training. For indoor scene point cloud, PointContrast [11] first reconstructs the whole scene and uses contrastive learning for pre-training. The research thread was followed by P4Contrast [13] and Contrastive-Scene-Context [12]. For outdoor scene LiDAR point clouds, there exists two primary schools of thought, depending on whether labels are required during the pre-training stage. The first school takes a semi-supervised 3D pre-training approach by utilizing a small sest of labels during pre-training, where the pre-training tasks include object detection, occupancy prediction (e.g., AD-PT [31] and SPOT [32]). The second takes an unsupervised 3D representation learning approach, where no label is required during pre-training: 1) Contrastive-based methods [16, 14, 33, 15, 34] create alternative (augmented) views of outdoor scene LiDAR point cloud and learns the representation by contrastive learning. 2) Mask-Autoencoder-based methods [17, 18, 19, 20, 21, 35] mask the input LiDAR point clouds and reconstruct the masked elements as the supervision signal. Following these lines of work, [14, 33] utilizes adjacent frames of LiDAR point clouds as views for contrastive learning. However, as the scenes are dynamic and there are no labels, the positive and negative pairs selection is very noisy [16], resulting in pre-training performance degradation. T-MAE [35] proposes to use the adjacent previous frame of LiDAR point clouds for masked autoencoding pre-training, but temporal information is limited to two frames (less than 0.5 second) and only history information is used. Also, ego action is not modeled in [35], which fails to learn the interaction between ego-vehicle and other traffic participants. Furthermore, the decoder in [35] is simply Multi-layer Perceptron on occupied 3D space and neglects empty parts of the scenes, which also matters in downstream tasks. For RGB image pretraining, ViDAR [36] utilizes future point clouds to pre-train image encoders with occupancy-based decoder, but suffers from the similar problem of [35]. Additionally, ViDAR also neglects current LiDAR point clouds, which is important for downstream 3D perception tasks. Besides, some works leverage 2D image prior to pre-train LiDAR encoder [37, 38, 39]. Unlike them, we propose TREND and use temporal forecasting as the pre-training goal. TREND utilizes a Recurrent Embedding scheme to integrate ego actions for temporal 3D embeddings and a Temporal LiDAR Neural Field as decoder to render both current and future point clouds for pre-training.

**Neural Field** plays an important role in 3D scene representation [25, 26]. IAE [40] and Ponder [19, 20] are pioneering work to use neural field in 3D pre-training and both use reconstruction as pre-training task; UniPAD [21] extends this line of work. The neural fields in [25, 26, 19, 20, 21] are originally designed for camera modality, which neglects characteristic of LiDAR point clouds and temporal information. Unlike them, we explore time-dependent neural field for LiDAR geometry and intensity by proposing a novel pre-training decoder and task to forecast future LiDAR point clouds.

**Scene Flow and LiDAR Forecasting.** 3D scene flow [41, 42, 43, 44, 45, 46, 47, 48, 49] has long been explored. Given current and past point clouds, the goal is to estimate per-point translation for the current point clouds. LiDAR forecasting take past and current observations as inputs and predict the future LiDAR point clouds, which necessitates the induction that we hypothesize beneficial for downstream perception tasks. Representative works include 4DOCC [22], Copilot4D [24] and Uno [23]. 4DOCC [22] uses a U-Net convolutional architecture and conduct differentiable rendering on the BEV feature map to predict the LiDAR observation in the future. Copilot4D [24] first trains a tokenizer/encoder for LiDAR point cloud with masked-and-reconstruction task and then freeze the encoder to train a diffusion-based decoder for LiDAR forecasting. Uno [23] proposes to use occupancy field as the scene representation for point cloud forecasting. The forecasting training stage in Copilot4D [24] does not envolve the 3D encoder for LiDAR point cloud and only focuses on training the diffusion-based decoder, which actually does not introduce temporal information into the 3D encoder. 4DOCC [22] and Uno [23] train the 3D encoder for forecasting but do not take the action of the autonomous vehicle into consideration. However, the interaction between the autonomous vehicle and the traffic participants is important for the prediction. The above methods study treat forecasting as the primary perception task. Unlike them, TREND adopts point cloud forecasting for unsupervised 3D representation learning and aims to improve downstream perception tasks via pre-training. TREND incorporate ego action, which previous works do not account, for pre-training.

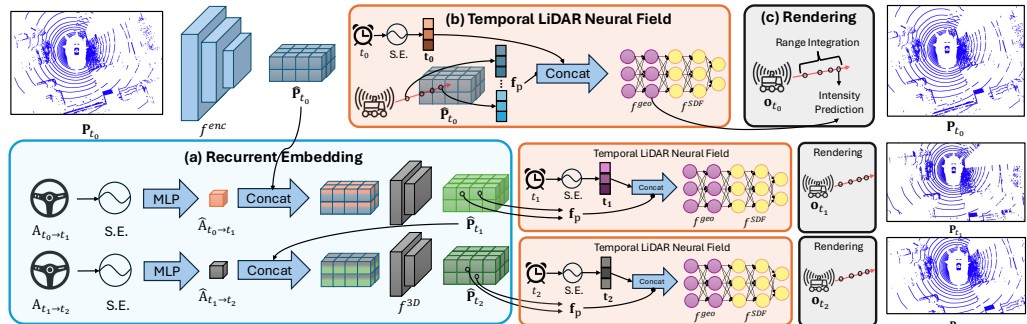

Figure 2: The overview of TREND. TREND first uses "S.E." (sinusoidal encoding [58, 59]) and Multi-layer Perceptron to embed ego actions and concatenate them with previous 3D embeddings to generate 3D embeddings at different timestamps (a). Then the Temporal LiDAR Neural Field (b) is used to represent the temporal 3D scene. The queried features, timestamp embeddings and point positions are concatenated and fed into geometry feature function $f^{\text{geo}}$. Next, we separately predict intensity values and signed distance values with geometry features at sampled points and conduct differentiable rendering to reconstruct and forecast LiDAR point clouds for pre-training.

**LiDAR-based 3D Perception.** LiDAR 3D object detection aims to take the raw LiDAR point clouds as input and predict bounding boxes for different object categories in the scene. Existing literature on LiDAR-based 3D object detection can be divided into three main streams based on the 3D encoder. 1) Point-based methods [50, 51] apply point-level embedding to detect objects in the 3D space. 2) Embraced by [1, 2, 6, 5], voxel-based methods apply voxelization to the raw point clouds and use sparse 3D convolution to encode the 3D voxels. 3) Point-voxel-combination methods [3, 4] combine the point-level and voxel-level features from 1) and 2). LiDAR semantic segmentation predicts category label for each LiDAR point. Cylinder3D [52], PVKD [53], Point-Transformer [54, 55, 56] and SphereFormer [57] achieve excellent performance for LiDAR segmentation task. In this paper, we use both LiDAR 3D object detection and segmentation as downstream tasks and fine-tune on various datasets [27, 28, 29, 30] to evaluate the effectiveness of TREND.

## 3 Method

In this section, we introduce TREND for unsupervised 3D representation learning on LiDAR perception via temporal forecasting. Fig. 2 is an overview of TREND. To overcome the two challenges in incorporating temporal forecasting for unsupervised 3D representation learning, we propose (a) the Recurrent Embedding scheme that accounts for the effect of autonomous vehicle's (ego)action to generate 3D embeddings at different timestamps, (b) Temporal LiDAR Neural Field, which represents the 3D scene by the geometry function and signed distance value function. The pre-training goal is to render current and future point clouds to compute loss and optimize the network. We first introduce problem formulation and overall pipeline in Section 3.1. Then we describe the Recurrent Embedding scheme and the Temporal Neural Field in details respectively in Section 3.2 and 3.3. Finally in Section 3.4, we discuss the differentiable rendering process and loss computation.

### 3.1 Problem Formulation and Overview

**Notations.** To start with, LiDAR point clouds are denoted as $\mathbf{P} = [\mathbf{L}, \mathbf{F}] \in \mathbb{R}^{N \times (3+d)}$, the concatenation of the $xyz$-location $\mathbf{L} \in \mathbb{R}^{N \times 3}$ and point features $\mathbf{F} \in \mathbb{R}^{N \times d}$. Here $N$ denotes the number of points in the point clouds and $d$ the number of feature channels. For instance, $d = 1$ in Once [27] representing intensity and for Waymo [29], $d = 2$ are intensity and elongation. To indicate point clouds at different timestamps, we use subscripts and $\mathbf{P}_t = [\mathbf{L}_t, \mathbf{F}_t] \in \mathbb{R}^{N_t \times (3+d)}$ is point cloud at time $t \in \{t_0, t_1, t_2, ..., t_k\}$, where $t_0$ indicates current timestamp and $t_1, t_2, ...t_k$ are future timestamps. At each timestamp $t_n$, we also have the action $\mathbf{A}_{t_n \to t_{n+1}} = [\Delta_x, \Delta_y, \Delta_\theta] \in \mathbb{R}^3$ of the autonomous vehicle and it is described with the relative translation on x-y plane $(\Delta_x, \Delta_y)$ and orientation with respect to z-axis $(\Delta_\theta)$ between timestamp $t_n$ and $t_{n+1}$.

**Overview.** Our goal is to pre-train the 3D encoder $f^{\text{enc}}$ in an unsupervised manner via forecasting. We begin by embedding $\mathbf{P}_{t_0}$ with the 3D encoder $f^{\text{enc}}$ to obtain the 3D representations

$$\hat{\mathbf{P}}_{t_0} = f^{\text{enc}}(\mathbf{P}_{t_0}), \tag{1}$$

where $\hat{\mathbf{P}}_{t_0} \in \mathbb{R}^{D \times H \times W \times \hat{d}}$ denotes the embedded 3D features with spatial resolution of $D \times H \times W$ and $\hat{d}$ feature channels. Then with $\hat{\mathbf{P}}_{t_0}$ and action at different timestamps $\mathbf{A}_{t_n \to t_{n+1}}$ as inputs, we apply the recurrent embedding scheme $f^{\text{rec}}$ and get the 3D embedding at different timestamps

$$\hat{\mathbf{P}}_{t_{n+1}} = f^{\text{rec}}(\mathbf{A}_{t_n \to t_{n+1}}, \hat{\mathbf{P}}_{t_n}), \tag{2}$$

where $n = 0, 1, \ldots$. Finally, to guide the training of 3D encoder in an unsupervised manner, we use a Temporal Neural Field to reconstruct and forecast LiDAR point clouds $\tilde{\mathbf{P}}_{t_n}$

$$\tilde{\mathbf{P}}_{t_n} = f^{\text{render}}(\hat{\mathbf{P}}_{t_n}), \tag{3}$$

and compute the loss against the raw observation $\mathbf{P}_{t_n}$ for optimization. Note that all the LiDAR point clouds are transformed into the coordinate frame of $t_0$ for consistency.

## 3.2 Recurrent Embedding Scheme

In order to introduce temporal information into 3D pre-training for $f^{\text{enc}}$, we embed the 3D representation $\mathbf{P}_{t_0}$ of the current frame at $t_0$ into future 3D representation ($\mathbf{P}_{t_1}$, $\mathbf{P}_{t_2}$ ...). To achieve this, previous literature [22, 23] directly apply learnable 3D/2D decoders but neglect the effect of autonomous vehicle's action $\mathbf{A}_{t_n \to t_{n+1}}$. However, the action of the autonomous vehicle is a part of the interaction between the ego-vehicle and traffic participants, and may influence the motion of the traffic participants on the road; hence, serving as a predictor. For example, if the autonomous vehicle does not move for some time, other traffic participants might move faster and vice versa. Thus, we propose to take $\mathbf{A}_{t_n \to t_{n+1}}$ into account and use a recurrent embedding scheme.

To begin, sinusoidal encoding [58, 59] are used to encode the relative translational components $[\Delta_x, \Delta_y]$ in raw action $\mathbf{A}_{t_n \to t_{n+1}}$ with sinusoidal functions of different frequencies. The resulting translation feature $\mathbf{f}_{\text{tl}} \in \mathbb{R}^{d_{\sin}}$ contains $d_{\sin}$ bounded scalars. Then we use $\mathbf{f}_{\text{rot}} = [\sin \Delta_\theta, \cos \Delta_\theta] \in \mathbb{R}^2$ to represent the rotational component in $\mathbf{A}_{t_n \to t_{n+1}}$ and concatenate both features to generate an initial action embedding $\tilde{\mathbf{A}}_{t_n \to t_{n+1}} = [\mathbf{f}_{\text{tl}}, \mathbf{f}_{\text{rot}}] \in \mathbb{R}^{d_{\sin}+2}$ for $\mathbf{A}_{t_n \to t_{n+1}}$. Note, this initial embedding process does not require any learnable parameter. To then further learn to embed $\tilde{\mathbf{A}}_{t_n \to t_{n+1}}$, we apply a shared shallow multi-layer perceptron (MLP) $f^{\text{act}}$ and project it to $\hat{\mathbf{A}}_{t_n \to t_{n+1}} \in \mathbb{R}^{d_{\text{act}}}$

$$\hat{\mathbf{A}}_{t_n \to t_{n+1}} = f^{\text{act}}(\tilde{\mathbf{A}}_{t_n \to t_{n+1}}). \tag{4}$$

With 3D embeddings at current timestamp $\hat{\mathbf{P}}_{t_0}$ and action embeddings at future timestamps $\hat{\mathbf{A}}_{t_n \to t_{n+1}}$, we broadcast $\hat{\mathbf{A}}_{t_n \to t_{n+1}}$ to the shape of $\hat{\mathbf{P}}_{t_n}$ and concatenate it with $\hat{\mathbf{P}}_{t_n}$ along the feature dimension, followed by a shared shallow 3D dense convolution $f^{\text{3D}}$ to get the embedding at different timestamps $\hat{\mathbf{P}}_{t_{n+1}} \in \mathbb{R}^{D \times H \times W \times \hat{d}}$.

$$\hat{\mathbf{P}}_{t_{n+1}} = f^{\text{3D}}([\mathbf{A}_{t_n \to t_{n+1}}, \hat{\mathbf{P}}_{t_n}]), \ n = 0, 1, \ldots \tag{5}$$

While local features reflect the understanding of other traffic participants and the environment, the concatenation provides local features with understanding of ego-vehicle motion. Despite the feature vector containing the vehicle ego-motion, the remainder of the feature vector allows us to predict the feature evolution. This recurrent embedding scheme allows us to model the evolution of the latent scene features based on vehicle ego-motion.

## 3.3 Temporal LiDAR Neural Field

We propose to use Neural field to represent the 3D scene around the autonomous vehicle at different timestamp $t$, which is the basis for LiDAR point clouds rendering. Previous work [25, 26, 60, 61, 62] design neural field for image modality and neglect both LiDAR characteristic and temporal information. On the contrary, we propose Temporal LiDAR Neural Field. As shown in Fig. 2, the goal of Temporal LiDAR Neural Field is to infer the geometry features and the signed distance

value [63, 64] for a point $\mathbf{p}$ in 3D space at timestamp $t$. Given the location of a specific point $\mathbf{p} = [x, y, z] \in \mathbb{R}^3$ at timestamp $t$, we first query the feature $\mathbf{f}_p \in \mathbb{R}^{\hat{d}}$ at $\mathbf{p}$ with $\hat{\mathbf{P}}_t$ by trilinear interpolation $f^{\mathrm{tri}}$ implemented in Pytorch [65]:

$$\mathbf{f}_p = f^{\mathrm{tri}}(\mathbf{p}, \hat{\mathbf{P}}_t). \tag{6}$$

Similar to initial action embedding in Section 3.2, we apply sinusoidal encoding [58, 59] to encode timestamp $t$ to $\mathbf{f}_t \in \mathbb{R}^{d_{\mathrm{sin}}}$. Taking the concatenation of location $\mathbf{p}$, $\mathbf{f}_t$ and the queried feature $\mathbf{f}_p$ as inputs, we first predict the geometry features $\mathbf{f}_{\mathrm{geo}} \in \mathbb{R}^{d_{\mathrm{geo}}}$ with $f^{\mathrm{geo}}$ and then the signed distance value $s \in \mathbb{R}$ [63, 64] with $f^{\mathrm{SDF}}$, which are parameterized by Multi-layer Perceptron:

$$\mathbf{f}_{\mathrm{geo}} = f^{\mathrm{geo}}([\mathbf{p}, \mathbf{t}, \mathbf{f}]) \quad ; \quad s = f^{\mathrm{SDF}}(\mathbf{f}_{\mathrm{geo}}). \tag{7}$$

### 3.4 Point Cloud Rendering

Each LiDAR point $\mathbf{p}$ can described by the sensor origin $\mathbf{o} \in \mathbb{R}^3$, normalized direction $\mathbf{d} \in \mathbb{R}^3$, and the range $r \in \mathbb{R}$, i.e., $\mathbf{p} = o + r\mathbf{d}$. Similar to [25, 26, 60, 61, 62], we first sample $N_{\mathrm{render}}$ rays at the sensor position $\mathbf{o}$ that trave along the normalized direction $\mathbf{d}$, and apply differentiable rendering to predict the range of LiDAR beam rays at different timestamp $t \in \{t_0, t_1, t_2, ...\}$ with our Temporal Neural Field.

**Sampling of $N_{\mathrm{render}}$.** LiDAR points on the ground are less informative and we filter out ground points by setting a threshold $z_{\mathrm{thd}}$ for $z$ values of the point position in vehicle coordinate frame. $z_{\mathrm{thd}}$ is determined by sensor height provided in the datasets. After that, we uniformly sample $N_{\mathrm{render}}$ at timestamp $t_n$ to conduct range rendering and loss computation.

**Range Rendering.** For a specific timestamp $t$, we sample $N_{\mathrm{ray}}$ points following [26] along each ray and construct the point set $\{\mathbf{p}_n = \mathbf{o} + r_n\mathbf{d}\}_{n=1}^{N_{\mathrm{ray}}}$. For each point in the point set, we estimate the signed distance value $s_n$ as described in Section 3.3. Then we predict the occupancy value $\alpha_n$

$$\alpha_n = \max\left(\frac{\Phi_z(s_n) - \Phi_z(s_{n+1})}{\Phi_z(s_n)}, 0\right), \tag{8}$$

where $\Phi_z(x) = (1 + e^{-zx})^{-1}$ is the sigmoid function with a learnable scalar $z$. With $\alpha_n$, we estimate the accumulated transmittance $\mathcal{T}_n$ [26] by $\mathcal{T}_n = \prod_{i=1}^{n-1}(1 - \alpha_i)$. We follow conventional rendering methods [26] to compute an occlusion-aware and unbiased weight $w_n = \mathcal{T}_n\alpha_n$. Differentiable rendering is conducted by integrating sampled points along the ray, leading to the predicted range $\tilde{r}$,

$$\tilde{r} = \sum_{n=1}^{N_{\mathrm{ray}}} w_n * r_n. \tag{9}$$

**Intensity Prediction.** According to [66], the intensity of LiDAR point clouds is decided by three factors: sensor system, surface material, and injection angle. Moreover, injection angle can be inferred by the ray direction $\mathbf{d}$ and surface normal. The geometry feature $\mathbf{f}_{\mathrm{geo}}$ and queried feature $\mathbf{f}_p$ at the scanned point includes information about the surface normal and material respectively. Thus we first embed the ray direction $\mathbf{d}$ by a Multi-layer Perceptron $f^{\mathrm{dir}}$. Then we concatenate the direction embedding $\mathbf{f}_{\mathrm{dir}} \in \mathbb{R}^{d_{\mathrm{dir}}}$, geometry feature $\mathbf{f}_{\mathrm{geo}}$ and queried feature $\mathbf{f}_p$ at the scanned point and apply an intensity network $f^{\mathrm{int}}$ to predict the intensity $\tilde{\mathcal{I}}$

$$\tilde{\mathcal{I}} = f^{\mathrm{int}}([\mathbf{f}_{\mathrm{dir}}, \mathbf{f}_{\mathrm{geo}}, \mathbf{f}_p]). \tag{10}$$

**Loss Function.** For each sampled ray, we have the observed range $r^i$ and intensity $\mathcal{I}^i$ and the predicted ones $\tilde{r}^i$ and $\tilde{\mathcal{I}}^i$, with which we compute an L1 loss; meanwhile, the expected signed distance value of the observed points $s_i$ is zero. We integrate this constraint into the loss function.

$$\mathcal{L}_{t_n} = \frac{1}{N_{\mathrm{render}}} \sum_{i=1}^{N_{\mathrm{render}}} (|r^i - \tilde{r}^i| + |\mathcal{I}^i - \tilde{\mathcal{I}}^i| + |s_i|). \tag{11}$$

## 3.5 Curriculum Learning for Forecasting Length

It is difficult for a randomly initialized network to directly learn to forecast several frames of LiDAR point clouds. Thus we propose to borrow the idea of curriculum learning [67, 68] and gradually increase the forecasting length. Specifically, we optimize the network with $N_{\text{curri}}^l$ curriculum learning epochs for $\{\mathbf{P}_{t_n}\}_{n=0}^l$, where $l = 1, 2, ....$ Because the observation nearer to current timestamp introduce more information about the current stage, we always reconstruct the current LiDAR point clouds and apply a decay weights $p(m)$ $(m = 1, 2, ..., l)$ to sample a future timestamp, where $p(m) > p(m+1)$ always holds. The final loss is computed as,

$$\mathcal{L} = \mathcal{L}_{t_0} + \mathcal{L}_{t_m}, \ \ m \sim p(m). \tag{12}$$

## 3.6 Discussions

**Theoretical Insight of TREND.** We provide an analysis in the aspect of information theory [69] and minimal sufficient representation [70, 71, 72]. Let data be $\mathbf{X}$, its representation be $\mathbf{Z}$ and a downstream task be $\mathbf{Y}$. $\mathbf{Z}$ is sufficient for $\mathbf{Y}$ if it is faithful to the task, e.g., fidelity of predictions. However, one may choose a $\mathbf{Z}$, including $\mathbf{X}$ – by the definition of data processing inequality, if $\mathbf{X}$ is sufficient, then $\mathbf{Z}$ is also sufficient. According to our discussion in Section 1, there are factors (nuisances) in the data that (negatively) impact predictions, and has implications towards generalization. Hence, it is desirable for a representation to be minimal, that is, containing the smallest amount of information, but sufficient for $\mathbf{Y}$. The instantiation of this is the Information Bottleneck (IB) Lagrangian:

$$\max \ I(\mathbf{Z}; \mathbf{Y}) - \beta I(\mathbf{X}; \mathbf{Z}), \tag{13}$$

where $I(;)$ denotes the mutual information between two random variables. Maximizing IB Lagrangian leads to fidelity for the task through the first data term and minimality or compression through the second bottleneck term. Naturally, $\beta$ controls the compression, where larger compression discards nuisance variability. The nuisance $\mathbf{N}$ influence $\mathbf{Z}$ only through $\mathbf{X}$, which follows the casual chain $\mathbf{N} \rightarrow \mathbf{X} \rightarrow \mathbf{Z}$. Thus we have $I(\mathbf{Z}; \mathbf{N}) \leq I(\mathbf{X}; \mathbf{Z}) - I(\mathbf{Z}; \mathbf{Y})$. Hence, the relationship between IB Langrangian and our proposal of temporal forecasting as a mechanism for unsupervised representation learning lies in the choice of modeling nuisance variables $\mathbf{N}$. What we want to accomplish is to minimize $I(\mathbf{Z}; \mathbf{N})$. We posit that the temporal dynamics within a dataset better exhibit the set of nuisance variables than does a handcrafted set through data augmentation. While it is intractable to quantify $I(\mathbf{Z}; \mathbf{N})$ directly, our empirical findings suggest that representations learned through temporal forecasting better suppress nuisances and improve downstream performance.

**Memory and Computational Overhead of TREND.** While TREND introduces temporal forecasting and neural field rendering, the actual memory costs are comparable to baseline methods. We utilize two design choices when sampling the rendering rays: ground point filtering and uniform ray sampling to make the GPU memory consumption feasible for TREND. In our experiments, all pre-training methods utilize the same GPU memory. As for computational cost during pre-training, since TREND employs Recurrent Embedding scheme, TREND requires approximately 8% more time than previous methods per epoch (65 mins v.s. 60 mins on 8-A100) for pre-training. Besides, recurrent embedding and temporal neural fields are not used during both fine-tuning and inference. The downstream model architecture, computational cost, and memory usage are identical across all methods.

## 4 Experiments

Unsupervised 3D representation learning aims to pre-train 3D backbones and use the pre-trained weights to initialize downstream models for performance improvement. In this section, we design experiments to demonstrate the effectiveness of TREND as compared to previous methods. We start with introducing experiment settings in Section 4.1. Then main results are provided in Section 4.2. Finally, additional experiment results and ablation study are discussed in Section 4.3.

### 4.1 Experiment Settings

**Datasets.** We conduct experiments on four popular autonomous driving datasets including Once [27] NuScenes [28], Waymo [29] and SemanticKITTI [30]. Once utilizes a 40-beam LiDAR to collect 1 million LiDAR frames and labels 15k of them. Due to the computation resource limitation, we conduct pre-training with TREND on the small split of the unlabeled data (100k frames) and fine-tune

| Init. | F.T. | mAP | Vehicle | | | Pedestrian | | | Cyclist | | |
|---|---|---|---|---|---|---|---|---|---|---|---|
| | | | 0-30m | 30-50m | 50m- | 0-30m | 30-50m | 50m- | 0-30m | 30-50m | 50m- |
| Ran. | | 46.07 | 76.71 | 51.15 | 31.84 | 37.53 | 20.12 | 9.84 | 62.00 | 42.61 | 24.18 |
| [73] | | $44.69^{-1.38}$ | 74.04 | 49.66 | 29.63 | 33.98 | 20.94 | 12.42 | 60.63 | 43.14 | 23.63 |
| [74] | | $44.43^{-1.64}$ | 76.52 | 49.48 | 30.18 | 35.32 | 18.96 | 9.36 | 60.47 | 40.94 | 22.99 |
| [22] | 5% | $40.84^{-5.23}$ | 74.23 | 46.64 | 29.45 | 29.85 | 17.31 | 9.56 | 57.47 | 33.59 | 18.34 |
| [35] | | $45.12^{-0.95}$ | 74.20 | 49.52 | 30.25 | 37.51 | 20.46 | 9.97 | 60.93 | 41.82 | 25.75 |
| [21] | | $46.23^{+0.16}$ | 78.76 | 55.77 | 37.81 | 31.65 | 16.09 | 8.78 | 64.90 | 44.18 | 24.73 |
| Ours | | $\mathbf{47.84}^{+1.77}$ | 79.14 | 55.68 | 36.34 | 35.23 | 18.00 | 11.18 | 64.99 | 45.80 | 28.15 |
| Ran. | | 57.68 | 82.70 | 63.37 | 46.34 | 52.61 | 36.48 | 19.03 | 71.03 | 55.34 | 36.34 |
| [73] | | $56.27^{-1.41}$ | 81.01 | 61.13 | 43.63 | 49.78 | 35.51 | 20.02 | 69.55 | 52.58 | 34.94 |
| [74] | | $57.09^{-0.59}$ | 83.51 | 62.57 | 46.28 | 50.96 | 34.55 | 17.90 | 70.37 | 54.50 | 36.79 |
| [22] | 20% | $54.30^{-3.38}$ | 80.69 | 58.95 | 42.13 | 45.09 | 33.14 | 18.04 | 68.90 | 52.20 | 35.09 |
| [35] | | $57.23^{-0.45}$ | 81.66 | 62.64 | 45.14 | 51.32 | 34.80 | 17.26 | 70.87 | 54.08 | 33.25 |
| [21] | | $58.08^{+0.40}$ | 84.23 | 65.44 | 48.65 | 49.48 | 34.84 | 19.38 | 70.76 | 55.75 | 38.89 |
| Ours | | $\mathbf{58.93}^{+1.25}$ | 84.08 | 65.80 | 50.51 | 50.31 | 33.37 | 19.42 | 72.54 | 56.31 | 39.26 |
| Ran. | | 65.03 | 88.18 | 74.23 | 61.75 | 57.32 | 38.90 | 21.96 | 78.07 | 64.32 | 48.16 |
| [73] | | $64.19^{-0.84}$ | 86.07 | 72.44 | 59.28 | 57.25 | 37.14 | 22.25 | 77.62 | 61.94 | 45.91 |
| [74] | | $65.10^{+0.07}$ | 88.02 | 74.01 | 61.95 | 57.56 | 38.43 | 22.45 | 79.95 | 63.64 | 47.89 |
| [22] | 100% | $64.48^{-0.55}$ | 88.34 | 74.20 | 61.32 | 55.78 | 37.14 | 22.32 | 77.95 | 62.42 | 46.40 |
| [35] | | $65.25^{+0.22}$ | 88.31 | 72.67 | 62.87 | 57.48 | 39.55 | 24.30 | 77.92 | 63.07 | 48.34 |
| [21] | | $65.19^{+0.16}$ | 88.11 | 74.00 | 62.28 | 57.67 | 38.49 | 21.99 | 79.51 | 64.40 | 47.65 |
| Ours | | $\mathbf{66.09}^{+1.06}$ | 88.56 | 75.02 | 63.10 | 57.83 | 39.29 | 20.63 | 79.48 | 65.08 | 49.02 |

Table 1: Results on Once dataset [27]. "Init." indicates the initialization methods and "Ran." is random initialization. "F.T." is the ratio of sampled training data for fine-tuning stage. We show mAP for overall performance and APs for different categories within different ranges. Green color is used to highlight the performance improvement and red one for degradation. We also use bold font to highlight the best mAP at different fine-tuning ratio. All the results are in %.

the pre-trained backbone with the labeled training set. NuScenes uses a 32-beam LiDAR to collect 1000 scenes in Boston and Singapore, where 850 of them are used for training and the other 150 ones for validation. We use the whole training set without label for all the pre-training methods. Waymo equips the autonomous vehicle with one top 64-beam LiDAR and 4 corner LiDARs to collect point clouds. We use Waymo for evaluating the transferring ability of TREND. SemanticKITTI uses a 64-beam LiDAR for data collection and provides semantic labels for each point.

**Downstream Models and Evaluation Metrics.** We perform downstream 3D object detection task on Once [27], NuScenes [28] and Waymo [29] and LiDAR semantic segmentation task on SemanticKITTI [30]. We follow the implementations in the popular code repository called OpenPCDet [75] and select the SOTA models on different datasets. For Once and Waymo, we use CenterPoint [2] as the downstream model. For Once, Average precisions for different categories within different ranges (APs) and mean average precision (mAP) are used for evaluation. For Waymo, APs and APs with heading (APHs) computed at two difficulty levels (Level-1 and Level-2) are utilized. For NuScenes, we use Transfusion-LiDAR [6] as the downstream model. APs for different categories, mAP and NuScenes Detection Score (NDS) are used for evaluation. For SemanticKITTI, We use Cylinder3D [52] and Mean Intersection over Union (mIoU) and accuracy are computed.

**Downstream Training Setting.** The main goal of unsupervised 3D pre-training is to improve *sample efficiency instead of accelerating convergence*, which has been discussed in previous literature [76, 18]. Sample efficiency means the best performance we can achieve with the same model and the same number of labeled data. Thus, we first gradually increase the training iterations for randomly initialized models until convergence is observed, which means increasing number of training iterations does not further improve the performance. Then we fix the training iterations and use the same schedule for downstream fine-tuning with different pre-training methods.

**Baseline 3D Pre-training Methods.** We select five baseline methods. (1) ALSO [73], an occupancy-based method. (2) Occupancy-MAE [74], an masked-autoencoder method. (3) 4DOCC [22], a

| Init. | mAP | NDS | Car | Truck | Bus | Bar. | Mot. | Bic. | Ped. | T.C. |
|---|---|---|---|---|---|---|---|---|---|---|
| Ran. | 31.06 | 44.75 | 69.18 | 28.73 | 34.57 | 42.31 | 13.72 | 8.72 | 69.18 | 41.14 |
| [73] | $30.14^{-0.92}$ | $43.73^{-1.02}$ | 66.89 | 25.67 | 34.36 | 43.06 | 12.98 | 7.1 | 66.28 | 41.63 |
| [74] | $29.94^{-1.12}$ | $43.93^{-0.82}$ | 68.51 | 26.32 | 30.90 | 41.74 | 12.36 | 7.0 | 67.84 | 41.27 |
| [22] | $26.99^{-4.07}$ | $40.97^{-3.78}$ | 67.44 | 25.40 | 29.37 | 35.58 | 9.53 | 5.16 | 65.26 | 29.47 |
| [35] | $30.53^{-0.53}$ | $44.55^{-0.20}$ | 68.63 | 26.02 | 34.66 | 43.98 | 13.21 | 7.26 | 68.78 | 39.82 |
| [21] | $32.16^{+1.10}$ | $45.50^{+0.75}$ | 69.82 | 29.54 | 35.73 | 46.79 | 13.65 | 7.98 | 70.45 | 42.73 |
| Ours | $\mathbf{33.17}^{+2.11}$ | $\mathbf{46.21}^{+1.46}$ | 71.24 | 30.08 | 39.57 | 45.42 | 16.65 | 9.33 | 71.84 | 43.70 |

Table 2: Results on NuScenes [28] dataset. "Init." indicates the initialization methods and "Ran." is random initialization. We use green color to highlight performance improvement and red for degradation and bold fonts for best performance in mAP and NDS. All the results are in %.

| Init. | Level-1 | | Level-2 | | $\bar{\Delta}$ |
|---|---|---|---|---|---|
| | mAP | mAPH | mAP | mAPH | |
| Ran. | 61.60 | 58.58 | 55.62 | 52.87 | |
| [21] | 61.57 | 58.57 | 55.60 | 52.83 | **-0.03** |
| Ours | 62.32 | 59.22 | 56.37 | 53.84 | **+0.77** |

Table 3: Results for transferring experiments.

| Init. | mAP | Veh. | Ped. | Cyc. |
|---|---|---|---|---|
| Ran.* | 20.48 | 37.88 | 10.62 | 12.96 |
| [35]* | $21.58^{+1.10}$ | 37.83 | 11.72 | 15.19 |
| [21]* | $24.41^{+3.93}$ | 40.66 | 12.01 | 20.55 |
| Ours* | $29.95^{+9.47}$ | 44.99 | 16.28 | 28.59 |

Table 4: Results for accelerating convergence.

LiDAR point cloud forecasting method. (4) T-MAE [35], a concurrent work that utilizes previous adjacent frame of LiDAR point clouds for mased-and-reconstruction without considering action of the autonomous vehicle. (5) UniPAD [21], the masked-and-reconstruction-based method with rendering decoder. All the pre-trainings for baseline methods are conducted with their official code.

**Implementation.** For $f^{\text{enc}}$, we select the popular sparse convolution backbone [77]. The feature channels for embedded 3D feaures $\hat{\mathbf{P}}_{t_n}$, sinusoidal encoding and action embeddings are respectively set to $\hat{d} = 128$, $d_{\text{sin}} = 32$ and $d_{\text{act}} = 16$. The sampled ray number for rendering is $N_{\text{render}} = 12288$ and number of points along each ray is $N_{\text{ray}} = 48$. We set the pre-training learning rate as 0.0002 with a cosine learning schedule. **Random seed is fixed** for all pre-training and fine-tuning to guarantee reproducibility. More details can be found in Appendix A.

### 4.2 Main Results

**Results on Once Dataset.** We pre-train TREND and baseline methods on the small split of unlabeled data in Once and uniformly sample 5%, 20% and 100% of the labeled training set for downstream fine-tuning. The results are shown in Table 1. For overall performance, it can be found that TREND achieves the best performance across different ratio of fine-tuning data. The performance improvement compared to train-from-scratch model is 1.77, 1.25 and 1.06 respectively for 5%, 20% and 100% fine-tuning data, which is up to 4 times more than previous 3D unsupervised pre-training methods and demonstrates the effectiveness of TREND. As for different categories, TREND achieves up to 4% mAP improvement on Vehicle and Cyclist for 5% fine-tuning data and generally improve these two categories within different ranges. It can also be found that for Pedestrian class, TREND improves with 100% downstream data but degrades the performance a little bit under 5% and 20% fine-tuning data settings. We think this is because LiDAR point clouds stand for geometry and pedestrians are always captured in LiDAR point clouds with a cylinder-like shape, which is less-distinguishable as compared to cyclists and vehicle. For example, trash bins or poles also appear to be cylinder-like in LiDAR. Thus learning to reconstruct and forecast such less-distinguishable geometry harms the ability of the pre-trained backbone to identify pedestrians among similar cylinder-like shapes especially when there are less labeled downstream data, leading to a little degradation for 5% and 20% settings. Similar phenomenon is also observed for differentiable reconstruction method UniPAD.

**Results on NuScenes Dataset.** We pre-train TREND and baseline methods on the whole training set of NuScenes dataset. We then uniformly sample 175 frames of labeled LiDAR point clouds in the training set and conduct few-shot fine-tuning experiments. Results are shown in Table 2. Our proposed method TREND achieves 2.11% mAP and 1.46% NDS improvement over randomly initialization at convergence, which is the best among all the baselines. When compared to previous SOTA 3D pre-training method UniPAD, TREND achieves 91% more improvement for mAP and 94% more improvement for NDS. If we look into detailed categories, TREND achieves general

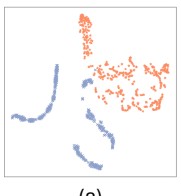 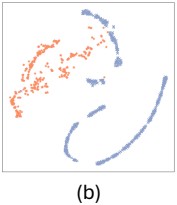 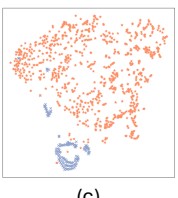 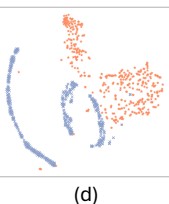

|        (a)         |        (b)         |        (c)         |        (d)         |

Figure 3: T-SNE visualization of TREND's features with Moving Object labels. The orange ones are static points while grey blue ones are moving points.

| Init. | mIoU | Acc |
| --- | --- | --- |
| Rand. | 28.23 | 70.68 |
| Occ-MAE | $27.82^{-0.41}$ | $76.57^{+5.89}$ |
| UniPAD | $29.98^{+1.75}$ | $76.10^{+5.42}$ |
| TREND | $31.12^{+2.89}$ | $79.82^{+9.14}$ |

Table 5: Results on SemanticKITTI [30].

| Rec. Emb. | N. F. | Tem. L. N. F. | mAP | NDS |
| --- | --- | --- | --- | --- |
| ✗ | ✗ | ✗ | 31.06 | 44.75 |
| ✗ | ✓ | ✗ | 32.16 | 45.26 |
| ✓ | ✓ | ✗ | 32.45 | 45.76 |
| ✓ | ✗ | ✓ | 33.17 | 46.21 |

Table 6: Results for ablation study.

improvement on different categories. For Car, Barrier, Motorcycle, Pedestrian and Traffic Cone, the improvement are more than 2% AP. For Bus, TREND introduce an improvement of 5% AP.

### 4.3 Other Results

**Transferring Experiments.** We use the backbone pre-trained on Once dataset to initialize Center-Point [2] and fine-tune the detector with 1% training data of Waymo [29]. The results are shown in Table 3. It can be found that TREND brings an average gain of 0.77 on mAPs and mAPHs, while UniPAD only achieves comparable performance. This demonstrates that TREND is able to pre-train the backbone on one dataset and then transfer to another dataset for performance improvement.

**Accelerating Convergence.** We use the default training iterations in OpenPCDet [75] to train $5\%$ Once data, which is the convergence acceleration setting and the experiment setting in most of the previous 3D pre-training literature. Results are shown in Table 4. It can be found that T-MAE and UniPAD accelerate convergence and TREND achieves the best performance.

**LiDAR Semantic Segmentation.** Results are show in Table 5. It can be found that TREND achieves best performance among different 3D pre-training baselines and improve the performance by 2.89% in mIoU and 9.14% in overall accuracy, demonstrating TREND's generalization ability across tasks.

**Ablation Study.** We conduct ablation study to analyze the contribution of different parts of TREND and results are shown in Table 6. "Rec. Emb.", "N. F." and "Tem. L. N. F." are respectively for Recurrent Embedding, Neural Field and Temporal LiDAR Neural Field. It can be found that using neural field for reconstruction pre-training brings little improvement and even degrades the NDS score compared to training-from-scratch. Adding Recurrent Embedding scheme with neural field improves the performance both on mAP and NDS, which demonstrates that Recurrent Embedding scheme is able to encode 3D features at different timestamps. Finally, with Temporal LiDAR Neural Field, TREND achieves the best performance both on mAP and NDS, showing that Temporal Neural Field better utilizes the temporal information in LiDAR sequence for unsupervised 3D pre-training.

**T-SNE for TREND's features.** We explored whether TREND's pre-trained features can distinguish between moving and static objects. We applied T-SNE [78] on TREND's features together with moving/static labels from [79]. In Figure 3, where static points are colored with orange and moving ones with grey blue, it can be found that although some noise exists, TREND's features for moving and static objects are generally separable after unsupervised pre-training.

## 5 Conclusion

In this paper, we propose TREND for unsupervised 3D representation learning via temporal forecasting, addressing the temporal embedding and scene representing challenges. With extensive experiments, we demonstrate that TREND is superior in improving downstream performance compared to previous SOTA techniques on various datasets and tasks. These results demonstrate the effectiveness of temporal forecasting in 3D pre-training. We believe TREND will facilitate our understanding on 3D perception in autonomous driving.

## Acknowledgments and Disclosure of Funding

This paper is supported by the general research fund of Hong Kong 17208825 and 17209324, and the Global Industrial Technology Cooperation Center (GITCC) through a grant agreement with the Korea Institute for Advancement of Technology (KIAT), project number P0028922.

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

## A  More Implementation Details.

During pre-training of TREND, we set the curriculum learning epoch as $N_{\text{curri}}^1 = 12$ and $N_{\text{curri}}^2 = 36$. We use mask augmentation for TREND with a masking rate of 0.9. The ego-motion (action) information is directly computed from the ego-vehicle poses provided in standard autonomous driving datasets (from IMU or GPS). All experiments are implemented with Pytorch framework. All pre-trainings are conducted on 8 A100 GPUs with batch size equals to 3 per GPU. All downstream tasks are trained on 4 A100 GPUs with default settings in OpenPCDet [75] except for training iterations. We will release code and pre-trained models.

## B  Repeated Evaluation.

We use the same random seed in all experiments in the main paper for reproducibility. As repeated evaluation can further reveal the training robustness, we further repeat the experiment on Once (20% downstream data) for 5 times and compute the mean and standard deviation of the results for randomly initialization, UniPAD and TREND, which are shown in Table 7. It can be found that TREND still achieves the best performance in mAP while largely reducing the standard deviation. This means pre-training by TREND alleviates the influence of random seed and makes training more stable.

| Init. | mAP | Vehicle | Pedestrian | Cyclist |
|---|---|---|---|---|
| Rand. | 57.29±0.29 | 68.99±0.10 | 43.29±0.93 | 59.59±0.23 |
| UniPAD | 58.00±0.32 | 71.78±0.16 | 41.78±0.70 | 60.44±0.48 |
| TREND | 58.74±0.11 | 73.07±0.16 | 42.02±0.39 | 61.12±0.22 |

Table 7: Results for repeated evaluation on Once dataset [27] with 20% downstream data. Mean and variance are in %.

## C  More Experiments on NuScenes

In this section, we conduct more fine-tuning experiments on NuScenes dataset. Specifically, we randomly sample 2.5% and 5% of NuScenes training set and train the randomly initialization model [6] until convergence is observed. Then we apply the pre-trained weight by TREND to initialize the model [6] and fine-tune it with the same training iterations. Results are shown in Table 8. It can be found that TREND consistently improve the performance in downstream 3D object detection task with different ratio of downstream training data.

| Init. | F.T. | mAP | NDS | Car | Truck | Bus | Barrier | Mot. | Bic. | Ped. | T.C. |
|---|---|---|---|---|---|---|---|---|---|---|---|
| Rand. | 2.5% | 45.35 | 55.36 | 76.74 | 40.89 | 50.07 | 57.48 | 41.58 | 26.13 | 76.67 | 55.77 |
| TREND | | $45.79^{+0.64}$ | $56.23^{+0.87}$ | 77.74 | 42.96 | 50.78 | 59.39 | 40.37 | 23.48 | 77.22 | 57.51 |
| Rand. | 5% | 51.56 | 60.24 | 80.22 | 48.56 | 58.69 | 63.42 | 50.84 | 36.59 | 79.29 | 60.30 |
| TREND | | $52.02^{+0.46}$ | $61.02^{+0.78}$ | 80.54 | 48.15 | 57.93 | 63.57 | 52.59 | 36.92 | 79.99 | 60.94 |

Table 8: Results for few shot fine-tuning on NuScenes [28] dataset. We randomly sample 2.5% and 5% of labeled point clouds in the training set and use Transfusion [6] as the downstream model for all the experiments here. Results of overall performance (mAP) and different categories (APs) are provided. "Init." indicates the initialization methods. "Rand" indicates the results where we gradually increase training iterations for train-from-scratch model until convergence is observed. Mot., Bic., Ped. and T.C. are abbreviations for Motorcycle, Bicycle, Pedestrian and Traffic Cone. We use green color to highlight the performance improvement brought by different initialization methods and bold fonts for best performance in mAP and NDS. All the results are in %.

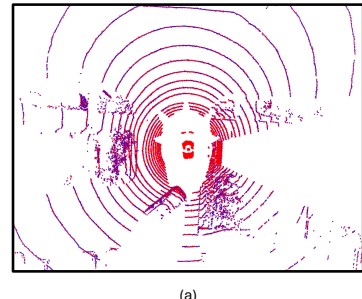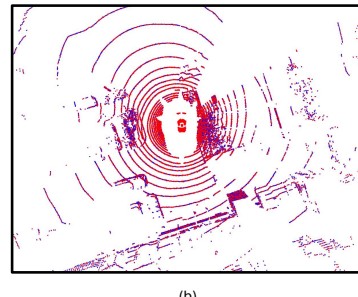

(a)          (b)

Figure 4: Forecasting Results. Blue points are raw observation from the dataset and red ones are prediction from TREND.

## D   Pre-training Decoder Choice Discussion.

There exists several other decoder choice including occupancy decoder [22, 32] and 3DGS (gaussian splatting) [80, 81, 82]. However, we consider neural-field decoder a more reasonable choice because the rendering process actively involve empty space into pre-training, of which information is actually crucial in LiDAR perception: a) In 3D perception, not only does the detectors need to detect where the objects are but also need to predict characteristics of objects (category, size, velocity and so on). Only use occupied space will harm the performance of 3D detectors. b) Almost all of the current SOTA 3D detectors [1, 2, 3, 4, 5, 6, 7] first generate a dense BEV-view feature map and predict the bounding boxes and object characteristics based on the dense feature map, which describes both occupied and empty space. c) Neural fields models signed distance values and model the entire scene including occupied and empty space, capturing the relationship between points and their surrounding space. This is significant for understanding scene geometry and potential object trajectories, providing a complete understanding of the environment. We further conduct experiments with 3DGS, occupancy, Copilot4D [24] and ViDAR [36] decoder on NuScenes. Downstream results of mAPs are 31.06% (random init), 30.84% (occupancy), 31.90% (GS-TREND), 31.08% (Copilot4D), 30.76% (ViDAR) and 33.17% (TREND). It can be found that replacing neural field in TREND with 3DGS or occupancy decoder degrades the representation quality.

## E   More Visualizations

**Forecasting results of TREND.** Qualitative visualization would enhance understanding of how TREND learns object motions. We first generate a visualization of forecasting results from TREND in Figure 4, where blue points are observation from dataset and red ones are prediction from TREND. As the forecasting error is small, it can be found that the difference is hard to be observed on the figure. Furthermore, we compute Mean Square Error on range prediction. Furthermore, we compute this error for moving objects and static objects respectively. Results (in meters) are: 0.0140. It can be found that the error is in centi-meter scale.

## F   Experiments on Hyper-parameter Sensitivity and Curriculum Learning.

**Forecasting Length.** In our experiments, we used a maximum forecast horizon of 4 frames (approximately 2 seconds). We conduct experiments on shorter and longer horizons on Once with 100% downstream labels. Results are 65.65% (3 frames), 66.09% (4 frames) and 65.33% (5 frames). We observe that representation quality first increases and then decreases when we add more timestamps for forecasting. First of all, it demonstrates that temporal information helps representation learning. Then, as longer sequence are not as predictable as shorter ones, the representation quality degrades after 2 seconds.

**Curriculum Learning Strategy.** The curriculum learning strategy is indeed important for TREND's performance. We conducted an ablation experiment training TREND without curriculum learning on Once dataset with 100% downstream labels. Result of mAP is 65.34%. It show that without

curriculum learning, performance of TREND drops by 0.75%, demonstrating that gradually increasing forecasting complexity is crucial for effective representation learning.

**Masking Rate.** The 90 % masking rate is determined empirically. We further conduct sensitivity study on masking rates on Nuscenes. Results of mAP are 32.56% for 80 percent masking rate, 33.17% for 90 percent, and 32.88% for 95 percent. While masking contributes to performance improvement, the temporal forecasting remains the primary driver of TREND's effectiveness.

**Sampling Strategy.** When sampling rays for rendering, we first filter ground LiDAR points (most of the ground points are background and less informative in pre-training) using the height of LiDAR sensor, which is originally provided in the datasets. Then we conduct uniform sampling. To investigate the influence of sampling strategy, we conduct ablation study of different sampling strategies including fully uniform sampling, farthest point sampling, uniform sampling with ground points filtering and farthest point sampling with ground points filtering. The experiments are conducted on NuScenes dataset. Results on mAP are 31.65% for FPS, 31.68% for uniform sampling, 32.52% for FPS with ground point filtering and 33.17% for uniform sampling with ground point filtering. It can be found that different sampling strategies make little difference but filtering ground points matters because ground points are background and less informative for the backbone pre-training.

## G   Comparison to pre-training method leveraging 2D images.

Leveraging 2D image priors to pre-traing LiDAR encoder also serves as a promising direction. Research efforts include SLiDR [37], LiMoE [38] and Sonota [39]. We further conduct experiments comparing these methods. As LiMoE([38])'s official repository only publish the first stage of its training, our experiment here utilize this part of code to pre-train the same LiDAR backbone we use. We also apply Sonota [39] to pre-train the same LiDAR backbone we use. Experiments are conducted on NuScenes dataset. Results are as belows:

| Init. | mAP | NDS |
|---|---|---|
| From-scratch | 31.06 | 44.75 |
| LiMoE (first stage) | 32.21 | 45.61 |
| Sonota | 32.32 | 46.07 |
| TREND | 33.17 | 46.21 |

It can be found that incorporating 2D prior yields similar results (a bit lower ; $< 1\%$ difference in mAP and NDS) as TREND, demonstrating the effectiveness of distillation from 2D prior. Meanwhile, as TREND only uses LiDAR modality for pre-training, it can be demonstrated that incorporating temporal information helps learn good 3D representations for downstream perception task. It would be a promising direction to bring both 2D prior and temporal information for pre-training.

## H   Broader Impact

This paper presents TREND, an unsupervised 3D representation learning method for LiDAR perception tasks in autonomous driving. There are three potential societal consequences of our work.

**Enhanced Safety and Robustness.** As experiment results show, TREND is able to improve performance on different tasks in autonomous driving, which enables autonomous vehicles (AVs) to better understand and adapt to complex environments without relying on extensive labeled datasets. This can lead to improved generalization across diverse road conditions, reducing the risk of accidents caused by unseen scenarios or edge cases.

**Environmental and Economic Benefits.** By reducing the reliance on manually annotated data, TREND lowers the computational and labor costs associated with dataset creation. Also, improved AV perception can lead to more energy-efficient driving behaviors, reducing fuel consumption.

**Job Displacement and Workforce Transition.** The adoption of unsupervised 3D pre-training in AVs could accelerate automation in the transportation sector, potentially displacing jobs in trucking, taxi services, and delivery industries.

# I   Limitations

While TREND demonstrates significant improvements over previous unsupervised 3D representation learning methods, two limitations should be acknowledged.

(1) Our approach shows varying effectiveness across different object classes. As observed in our experiments, TREND achieves substantial improvements for vehicle and cyclist classes but on Once dataset shows limited gains for pedestrian detection in low-data regimes. This is likely because pedestrians appear as cylinder-like shapes in LiDAR point clouds, making them less distinguishable from other similar structures in the environment.

(2) Our method currently focuses on the geometric aspects of temporal forecasting without explicitly modeling semantic. Incorporating semantic priors from other sensors like camera could potentially enhance the learned representations.

