# OpenReview forum: "TREND: Unsupervised 3D Representation Learning via Temporal Forecasting for LiDAR Perception"
_NeurIPS.cc/2025/Conference — NeurIPS 2025 spotlight_

### Official Review · Reviewer_jBSj · 2025-07-01

**Clarity:** 3
**Significance:** 3
**Originality:** 3
**Rating:** 5
**Confidence:** 4

**Summary:**

TREND proposed LiDAR pointcloud pretraining by 4D temporal forecasting-based unsupervised learning. With obtained 3D features and dynamic features, the TREND decode features by predict intensity and pointcloud by SDF-based rendering. With this pretraining, TREND shows the effectiveness on various datasets.

**Questions:**

- The pipeline should compute N times as the forecasting prediction, which may cause the inefficient computation. Should the motion feature divide with time feature and ego-motion? If not, it seems the voxel features from the inital timestep can be rendered in various timestep.
- There are some pretraining method by leveraging 2D images prior (i.e. LiMoE*, Sonata**). How much gap could be reduced by pointcloud only pretraining compared with image/pointcloud pretraining method?

LiMoE: Mixture of LiDAR Data Representation Learners from Automotive Scenes, CVPR'25
Sonata**: Self-Supervised Learning of Reliable Point Representations, CVPR'25

**Ethical Concerns:**

["NO or VERY MINOR ethics concerns only"]

**Final Justification:**

The authors have resolved my concerns regarding efficiency and comparisons with pre-training methods, so I will maintain my score as 5. I have also checked the other reviewers’ opinions, and the authors have addressed all concerns with additional comparisons.

**Limitations:**

yes, the authors already mentioned in supplements

**Paper Formatting Concerns:**

No concern for paper formatting

**Quality:**

3

**Strengths And Weaknesses:**

Strengths
- Novel temporal representation that integrated ego-motion into unsupervised pretraining.
- Simple temporal neural rendering with 3D sparse encoder.
- Lots of experiments with various dataset and ablations.

Weaknesses
- Lots of computations for neural rendering for each timestep.

---

> ### Author Rebuttal · Authors · 2025-07-29
>
> Dear Reviewer jBSj,
>
> Thank you for your precious time on the review and your constructive suggestions to improve our manuscript! We appreciate the acknowledgment that TREND is a novel and concise unsupervised 3D representation learning method and the experiments are extensive and comprehensive.
>
> We provide further discussions on your questions as belows:
>
> **Q1:Lots of computations for neural rendering for each timestep.**
>
> Thank you for this important concern. We provide detailed analysis divided into pre-training and fine-tuning phases:
>
> ***Pre-training Costs:*** While TREND introduces temporal forecasting and neural field rendering, the actual memory costs are comparable to baseline methods. All pre-training methods utilize approximately the same GPU memory as we utilize several design choices when sampling the rendering rays (Line 203-206 in the paper):
> (1) Ground point filtering: We remove less informative ground points using sensor height as thresholds, which is originally provided in the datasets.
> (2) Uniform ray sampling: After filtering, we uniformly sample $N_{render}=12,288$ rays per frame rather than processing all points.
>
> As for computational cost during pre-training, since TREND employs Recurrent Embedding scheme, TREND requires approximately 8% more time than previous methods per epoch (65 mins v.s. 60 mins on 8-A100) for pre-training, which is feasible.
>
> ***Fine-tuning (Inference) Costs:*** Recurrent embedding and temporal neural fields are only needed during pre-training stage. The downstream model architecture, computational cost, and memory usage are identical across all methods during both fine-tuning and inference. When we are fine-tuning for downstream perception tasks, we load the pre-trained 3D encoder weights to the same model architecture and train the same detector/segmentation model.
>
> This design ensures that while we leverage rich temporal information during pre-training, practical deployment remains efficient and scalable.
>
> **Q2: The pipeline should compute N times as the forecasting prediction, which may cause the inefficient computation. Should the motion feature divide with time feature and ego-motion? If not, it seems the voxel features from the inital timestep can be rendered in various timestep.**
>
> Thank you for this technical question. To clarify our approach:
>
> ***Feature evolution vs. rendering*** : The voxel features from the initial timestep (t0) cannot simply be rendered at various timesteps. In the recurrent embedding scheme, we first use Sinusoidal Encoding to embed ego-vehicle motion ($\Delta x$, $\Delta y$, $\Delta \theta$). Then we broadcast the ego-vehicle motion feature to each spatial location and concatenate it with local features. Next, a shallow 3D convolution (3 layers) is applied to project the features. While local features reflect the understanding of other traffic participants and the environment, this concatenation provides local features with understanding of ego-vehicle motion and allows us to predict the feature evolution, which is important for our forecasting pre-training goal.
>
> ***Computational efficiency***: While we compute features for N timesteps, each computation after the initial timestep is lightweight (shallow MLP + shallow 3D convolution). As we conduct a uniform sampling with filtering ground points (Line 203-206 in the paper), the forecasting rendering process is feasible. The only computational overhead is brought by recurrent embedding scheme, which requires approximately 8% more time than previous methods per epoch (65 mins v.s. 60 mins on 8-A100) during pre-training stage.
>
> The key insight is that simply reusing initial features would miss the temporal dynamics and the interaction between ego-vehicle and traffic participants (brought by ego-motion feature broadcasting and feature evolution) that make forecasting beneficial for representation learning.
>
> **Q3: There are some pretraining method by leveraging 2D images prior (i.e. LiMoE[A], Sonata[B] How much gap could be reduced by pointcloud only pretraining compared with image/pointcloud pretraining method?**
>
> Thanks for pointing it out! We will add discussion on pre-training method leveraging 2D images in the related works including LiMoE [A], Sonota [B] and SLiDR [C].  We further conduct experiments comparing methods distilling 2D image feature to LiDAR. As LiMoE's official repository currently only publish the first stage of its training, our experiment here utilize this part of code to pre-train the same LiDAR backbone we use. We also apply Sonota to pre-train the same LiDAR backbone we use. Experiments are conducted on NuScenes dataset. Results are as belows:
>
>
> | Init.        | mAP   | NDS   |
> |:------------:|:-----:|:-----:|
> | From-scratch | 31.06 | 44.75 |
> | LiMoE (first stage) | 32.21 | 45.61 |
> | Sonota | 32.32 | 46.07 |
> | TREND | 33.17 | 46.21 |
>
> It can be found that incorporating 2D prior yields similar results as TREND, demonstrating the effectiveness of distillation from 2D prior. Meanwhile, as TREND only uses LiDAR modality for pre-training, it can be demonstrated that incorporating temporal information helps learn good 3D representations for downstream perception task. It would be a promising direction to bring both 2D prior and temporal information for pre-training to further improve the performance.
>
> We will add the discussions and experiments here to the revision. Thank you again for your suggestions to improve our manuscript!
>
>
> [A] LiMoE: Mixture of LiDAR Data Representation Learners from Automotive Scenes, CVPR'25
>
> [B] Sonata: Self-Supervised Learning of Reliable Point Representations, CVPR'25
>
> [C] Image-to-lidar self-supervised distillation for autonomous driving CVPR'22.

---

> > ### Comment · Reviewer_jBSj · 2025-08-03
> >
> > Thank you for the great rebuttal. Since my original concerns have been addressed and additional experiments were provided,  especially for the comparison with point cloud pretraining methods, I will maintain my recommendation for acceptance.

---

> > > ### Author Response · Authors · 2025-08-03
> > >
> > > Dear Reviewer jBSj,
> > >
> > > Thank you for the reply and appreciation on our work and response. Thank you again for the constructive suggestions to improve the paper.

---

### Official Review · Reviewer_g95E · 2025-07-02

**Clarity:** 4
**Significance:** 4
**Originality:** 3
**Rating:** 5
**Confidence:** 5

**Summary:**

This paper proposes TREND, a novel unsupervised 3D representation learning method that uses temporal forecasting instead of traditional masked autoencoding or contrastive learning. The essential novelty involves predicting future LiDAR observations by incorporating ego-vehicle actions through a Recurrent Embedding scheme and using a Temporal LiDAR Neural Field specifically designed for LiDAR characteristics. The method is evaluated on multiple autonomous driving datasets for 3D object detection and semantic segmentation tasks.

**Questions:**

1. Please provide more reporting of absolute performance gains instead of the misleading "400% improvement" claim.
2. Why did you choose SDF over the other 3D representations, and have you conducted ablation studies on different architectural choices for the Temporal Neural Field?
3. Please provide computational cost analysis and theoretical insights into why temporal forecasting specifically helps representation learning.

**Ethical Concerns:**

["NO or VERY MINOR ethics concerns only"]

**Final Justification:**

The rebuttals resolved all my concerns regarding the technical selections and theoretical analysis, so I increased my score to 5.

**Limitations:**

Yes, the authors acknowledge limitations in Appendix H, specifically noting varying effectiveness across object classes and the focus on geometric rather than semantic aspects.

**Quality:**

4

**Strengths And Weaknesses:**

Strengths:
1. The paper addresses a key limitation in LiDAR unsupervised pre-training by transitioning from static pretext tasks to temporal forecasting, which potentially better captures dynamic scene properties relevant to downstream tasks.
2. The integration of ego-vehicle actions and the design of Temporal LiDAR Neural Field demonstrate good technical depth. Such a technique could be used to address fundamental challenges in temporal modeling for autonomous driving applications.
3. The authors conducted extensive experiments to demonstrate the effectiveness of the proposed methods, which effectively highlights the core contribution.
4. The writing is very clear, and the methodology is well-structured.

Major weaknesses:
1. The "400% improvement" claim is somehow misleading. The absolute improvement is limited. The description should be modified.
2. The motivation for the particular Temporal Neural Field architecture (e.g., the specific use of SDF, intensity modeling, and ray sampling strategy) could be strengthened by more direct comparisons or a clearer rationale. Why SDF over the other 3D representations? How sensitive is performance to different ray sampling strategies?
3. There is a lack of computational cost analysis, limited theoretical insight into why temporal forecasting aids representation learning, and insufficient explanation for class-specific performance variations.

---

> ### Author Rebuttal · Authors · 2025-07-29
>
> Dear Reviewer g95E,
>
> Thank you for your precious time on the review and your constructive suggestions to improve our manuscript! We appreciate the acknowledgment that TREND is a novel unsupervised 3D representation learning method, the experiments are extensive, the results show the effectiveness of TREND, the research addresses fundamental challenges in temporal modeling for autonomous driving applications and the writing is clear and well-structured.
>
> We provide further discussions on your questions as belows:
>
> **Q1: The claim on improvement.**
>
> While we understand that it would be better to incorporate absolute improvement in the claim, we respectfully disagree that our claim is misleading. The statement "TREND brings up to 400\% more improvement as compared to previous SOTA unsupervised 3D pre-training methods" is mathematically accurate.
>
> ***Why relative improvement comparison matters in pre-training***: In unsupervised representation learning, the key metric is improvement over random initialization, not absolute performance. For example, when previous best methods achieve +0.22\% improvement and TREND achieves +1.06\% improvement on Once's $100$% downstream experiments (Table 1 in the paper), TREND provides $\sim$ 4× (400\%) more improvement as compared to previous SOTA unsupervised 3D pre-training methods.
>
> ***Our absolute improvements are not limited***: We use a rigorous experimental setting where train-from-scratch models are trained to full convergence, making improvements harder to achieve but more meaningful and practical.
>
> (1) Our convergence setting (Tables 1-3): TREND achieves 1.77\% absolute improvement on Once, 2.11\% on NuScenes.
>
> (2) Previous early-stopping setting: (Table 4): TREND achieves 9.47\% absolute improvement, which is still the best among different initialization methods with a gap larger than 5\%.
>
> The convergence setting represents real-world deployment scenarios where practitioners train until performance plateaus. Our substantial improvements in this challenging setting demonstrate TREND's practical value.
>
> We will add the absolute statistic alongside the original claim to make it clearer. Thank you for this suggestion.
>
> **Q2: The motivation for the particular Temporal Neural Field architecture (e.g., the specific use of SDF, intensity modeling, and ray sampling strategy) could be strengthened by more direct comparisons or a clearer rationale. Why SDF over the other 3D representations?**
>
> As discussed in Appendix D (Line 587-603), we chose SDF over alternatives like occupancy-based decoders and Gaussian-Splatting ones because other decoders mainly focus on occupied space while rendering with SDF sample points along the rays and these points include both empty locations and occupied locations, which captures both occupied and empty space relationships. For perception tasks, empty space provides crucial semantic information (e.g., safe navigation corridors, object boundaries). The results in Appendix D (Line 587-603) validate this. Besides, we further conduct more experiments on occupancy-decoder-based method like ViDAR (we adapt it to LiDAR backbone pre-training, ViDAR$^*$) and Copilot4D on NuScenes dataset. Results are shown in the table below:
>
> | Init.        | mAP   | NDS   |
> |:------------:|:-----:|:-----:|
> | From-scratch | 31.06 | 44.75 |
> | Copilot4D | 31.08 | 45.05 |
> | ViDAR$^*$ | 30.76 | 44.52 |
> | TREND | 33.17 | 46.21 |
>
> The SDF-based neural field consistently outperforms occupancy decoders, validating our architectural choice.
>
> **Q3: How sensitive is performance to different ray sampling strategies?** Thank you for pointing this out. Currently we use a uniform sampling with height information (Line 203-206 in the paper). When sampling rays for rendering, we first filter ground LiDAR points (most of the ground points are background and less informative in pre-training) using the height of LiDAR sensor, which is originally provided in the datasets. Then we conduct uniform sampling. To further address the concern, we conduct ablation study of different sampling strategies including fully uniform sampling, farthest point sampling (FPS [A]), uniform sampling with ground points filtering and farthest point sampling with ground points filtering. The experiments are conducted on NuScenes dataset.
>
> | Sample Method        | mAP   | NDS   |
> |:------------:|:-----:|:-----:|
> | FPS | 31.65 | 45.42 |
> | Uniform | 31.68 | 45.77 |
> | FPS with ground filtering | 32.52 | 46.48 |
> | Uniform with ground filtering | 33.17 | 46.21 |
>
> It can be found that different sampling strategies make little difference but filtering ground points matters because ground points are background and less informative for the backbone pre-training.
>
> **Q4: Lack of computational cost analysis.**
>
> Thank you for this important concern. We provide detailed analysis divided into pre-training and fine-tuning phases:
>
> ***Pre-training Costs:*** While TREND introduces temporal forecasting and neural field rendering, the actual memory costs are comparable to baseline methods. All pre-training methods utilize approximately the same GPU memory as we utilize several design choices when sampling the rendering rays (Line 203-206 in the paper):
> (1) Ground point filtering: We remove less informative ground points using sensor height as thresholds, which is originally provided in the datasets.
> (2) Uniform ray sampling: After filtering, we uniformly sample $N_{render}=12,288$ rays per frame rather than processing all points.
>
> As for computational cost during pre-training, since TREND employs Recurrent Embedding scheme, TREND requires approximately 8% more time than previous methods per epoch (65 mins v.s. 60 mins on 8-A100) for pre-training, which is feasible.
>
> ***Fine-tuning (Inference) Costs:*** Recurrent embedding and temporal neural fields are only needed during pre-training stage. The downstream model architecture, computational cost, and memory usage are identical across all methods during both fine-tuning and inference. When we are fine-tuning for downstream perception tasks, we load the pre-trained 3D encoder weights to the same model architecture and train the same detector/segmentation model.
>
> This design ensures that while we leverage rich temporal information during pre-training, practical deployment remains efficient and scalable.
>
> **Q5: Theoretical insights into why temporal forecasting specifically helps representation learning.**
>
> Thank you for the suggestion. We provide a brief analysis in the aspect of information theory [B] and minimal sufficient representation [C,D,E] here and will add a detailed one in the revised appendix. For preliminaries: Let data be X, its representation be Z and a downstream task be Y. Z is sufficient for Y if it is faithful to the task, e.g., fidelity of predictions. However, one may choose a Z, including X -- by the definition of data processing inequality, if X is sufficient, then Z is also sufficient. As discussed in the main paper, there are factors (nuisances) in that data that (negatively) impact predictions, and has implications towards generalization. Hence, it is desirable for a representation to be minimal, e.g., containing the smallest amount of information, but sufficient for Y. The instantiation of this is the Information Bottleneck (IB) Lagrangian:
>
> $\max I(Z; Y) - \beta I(X; Z)$
>
> where I(;) denotes the mutual information between two random variables. Maximizing IB Lagrangian leads to fidelity for the task through the first data term and minimality or compression through the second bottleneck term. Naturally, $\beta$ controls the compression, where larger compression naturally discards nuisance variability.
>
> The nuisance N influence Z only through X, which follows the casual chain N --> X --> Z.
> Thus naturally,
> $I(Z; N) <= I(X; Z) - I(Z; Y)$
>
> Hence, the relationship between IB Langrangian and our proposal of temporal forecasting as a mechanism for unsupervised representation learning lies in the choice of modeling nuisance variables N. What we want to accomplish is to minimize $I(Z; N)$. We posit that the temporal dynamics within a dataset better exhibit the set of nuisance variables than does a handcrafted set through data augmentation. While it is intractable to quantify $I(Z; N)$ directly, our empirical findings suggest that representations learned through temporal forecasting better suppress nuisances and improve downstream performance.
>
> **Q6: insufficient explanation for class-specific performance variations.**
>
> Thank you for the suggestion. We observe that TREND shows limited performance for pedestrian detection in low-data regimes (Table 1, 5\% setting). This occurs because:
> (1) Pedestrians appear as cylinder-like shapes in LiDAR, similar to poles/trash bins.
> (2) Learning to reconstruct such geometrically simple shapes provides less discriminative information.
>
> It can also be found that this limitation diminishes with more training data (performance recovers at 100\% setting).
>
> While no links or pdf is allowed during rebuttal, in the revision, we will add this analysis and corresponding visualizations: (1) a zoom-in view of original Pedestrian point clouds and the rendered point clouds from TREND. (2) detection result visualization in low-data regimes (Table 1, 5\% setting).
>
> We will add the discussions and experiments here to the revision. Thank you again for your suggestions to improve our manuscript!
>
>
> [A] Moenning C, Dodgson N A. Fast marching farthest point sampling[R]. 2003.
>
> [B] Tishby N, Pereira F C, Bialek W. The information bottleneck method[J]. 2000.
>
> [C] Achille A, Soatto S. Emergence of invariance and disentanglement in deep representations[J]. Journal of Machine Learning Research, 2018.
>
> [D] Tsai Y H H, Wu Y, Salakhutdinov R, et al. Self-supervised learning from a multi-view perspective[J]. 2020.
>
> [E] Wang H, Guo X, Deng Z H, et al. Rethinking minimal sufficient representation in contrastive learning. CVPR' 2022

---

> > ### Comment · Reviewer_g95E · 2025-08-02
> >
> > I really appreciate your detailed responses to the reviews. I think it's a good rebuttal that clearly reclarified and exhibited the work's points and contributions further. Overall, it's good work and has clear contributions. I increased my score.

---

> > > ### Author Response · Authors · 2025-08-03
> > >
> > > Dear Reviewer g95E,
> > >
> > > Thank you for the comments and appreciation on our work and response. Thank you again for your constructive suggestions to improve the manuscript.

---

### Official Review · Reviewer_4GJv · 2025-07-03

**Clarity:** 3
**Significance:** 2
**Originality:** 2
**Rating:** 4
**Confidence:** 1

**Summary:**

This paper presents TREND (Temporal Rendering with Neural fielD), an unsupervised pretraining method for LiDAR-based 3D perception via temporal forecasting. TREND distinguishes itself from prior forecasting-based approaches by explicitly incorporating ego-motion into its temporal modeling using a Recurrent Embedding scheme. A Temporal LiDAR Neural Field is introduced to render future LiDAR frames for self-supervised learning. Experiments on four benchmark datasets demonstrate consistent improvements in 3D object detection and semantic segmentation compared to existing unsupervised pretraining methods.

**Questions:**

- Please consider revising the notation to better separate raw point clouds from feature embeddings and rendered outputs. This would significantly improve the clarity of the method section.
- Include more explicit comparisons or discussion with ViDAR and Copilot4D, especially to justify the added value of modeling ego-motion in the context of 3D pretraining.
- A brief qualitative discussion or visualization of failure cases (e.g., pedestrians, small objects) would strengthen the analysis.

**Ethical Concerns:**

["NO or VERY MINOR ethics concerns only"]

**Final Justification:**

After reading the rebuttal and the other reviews, my concerns on the novelty are resolved. Therefore, I maintain my positive rating.

**Limitations:**

Yes

**Quality:**

2

**Strengths And Weaknesses:**

**Strengths**
- Motivated contribution: Modeling temporal dynamics via forecasting aligns well with LiDAR-based autonomous driving data. Incorporating ego-motion is a meaningful step beyond prior work.
- Extensive evaluation: TREND is validated on multiple datasets and tasks, including transfer learning and segmentation, showing consistent improvements across the board.

**Weaknesses**
- Writing clarity: The method section suffers from inconsistent and overloaded notation. For example, the paper uses P, \hat{P}, and \tilde{P} to refer to raw point clouds, latent features, and rendered point clouds, respectively. This overload of the same base symbol makes the formulation difficult to follow. It would be clearer to use distinct symbols (e.g., F for features) to disambiguate between inputs and internal representations.
- Limited novelty: Although TREND is well-executed, the overall approach builds on existing temporal forecasting ideas. The key novelty lies in modeling ego-motion, which, while important, may not be sufficient to fully differentiate the method from recent concurrent work such as ViDAR or Copilot4D.

---

> ### Author Rebuttal · Authors · 2025-07-29
>
> Dear Reviewer 4GJv,
>
> Thank you for your thorough review and constructive feedback. We appreciate your recognition that our approach is well-motivated with extensive experiments showing consistent improvements. We address your concerns below:
>
> **Q1: Writing clarity.** Thank you for pointing it out. We will update the symbols accordingly to make it clearer and ensure consistent usage throughout. Meanwhile, we will further polish other clarification to strengthen the writing quality.
> (1) **P** for Raw LiDAR point clouds.
> (2) **F** for 3D feature embeddings from encoder.
> (3) **O** for rendering outputs.
>
> **Q2: About novelty and More comparison with ViDAR and Copilot4D.**
> We respectfully believe TREND offers substantial novelty beyond existing work:
>
> ***Primary contribution - Temporal forecasting for LiDAR-based unsupervised 3D representation learning.*** While forecasting has been studied for decades across various domains, TREND is the **first one** to systematically bring temporal forecasting into the realm of LiDAR-based unsupervised 3D representation learning. We aim to learn 3D representations for LiDAR perception (downstream tasks) instead of focusing on forecasting prediction results. Our experiments demonstrate that naive application of existing forecasting methods to representation learning yields minimal improvement or even degradation (Tables 1-2). This validates that bringing forecasting into unsupervised representation learning for LiDAR perception requires fundamental redesign, not just adaptation.
>
> ***Technical innovations enabling this new paradigm.***
>
> (1) Temporal LiDAR Neural Field: Unlike occupancy decoders in forecasting literature (ViDAR/Copillot4D), we apply neural field for the decoder. Furthermore, our proposed Temporal LiDAR Neural Field explicitly models time and LiDAR characteristics (intensity, geometry) for unsupervised 3D representation learning.
>
> (2) Ego-motion aware recurrent embedding: Incorporating ego-motion is important for modeling interaction between ego-vehicle and traffic participants.
>
> (3) Unified framework: Combines recurrent embedding with differentiable rendering to simultaneously learn spatial and temporal representations.
>
> ***Discussion on ViDAR and Copilot4D and Comparison.*** ViDAR is for image pre-training and only considers future forecasting, neglecting current state. Copilot4D focus on LiDAR forecasting accuracy, train the 3D backbone using only reconstruction objective and freeze the backbone when training a diffusion decoder to forecast future observation. This diffusion decoder cannot be simply used as part of the encoder. Thus the 3D backbone from Copilot4D only have the information about current state. However, for downstream perception tasks, both understanding about current state and future forecasting are important and TREND takes both into account. Besides, both of these two works utilize occupancy-based decoder, which only focuses on those occupied areas. For downstream perception tasks, not only does the occupied space matter but the empty ones also provide information. For example, the empty space around a pedestrian and a car should be different due to safety concern under different speed conditions. Also, empty space helps object boundary recognition. TREND samples points (including empty locations and occupies ones) along each ray and integrate to reconstruct and forecast, taking both into account and leading to better representations compared to ViDAR and Copillot4D. What's more, our Temporal LiDAR Neural Field introduce explicit timestamp and LiDAR intensity into the neural field, which further enhances the learned representations. Last but not least, incorporating ego-motion introduces the interaction between ego-vehicle and traffic participants into pre-training. This makes the representation better understand the surrounding environments and provides more benefit for downstream perception tasks, which is neglected by ViDAR.
>
> ***Experiment Validation.*** We adapt ViDAR for LiDAR backbone pre-training (ViDAR$^*$) and also use backbone pre-trained by Copilot4D for downstream fine-tuning on NuScenes dataset. We provide the results here.
>
>
> | Init.        | mAP   | NDS   |
> |:------------:|:-----:|:-----:|
> | From-scratch | 31.06 | 44.75 |
> | Copilot4D | 31.08 | 45.05 |
> | ViDAR$^*$ | 30.76 | 44.52 |
> | TREND | 33.17 | 46.21 |
>
> These results demonstrate that successfully bringing forecasting into unsupervised 3D representation learning requires careful technical design beyond existing forecasting approaches.
>
> **Q3: A brief qualitative discussion or visualization of failure cases (e.g., pedestrians, small objects) would strengthen the analysis.**
>
> Thank you for this suggestion. We observe that TREND shows limited performance for pedestrian detection in low-data regimes (Table 1, 5\% setting). This occurs because:
> (1) Pedestrians appear as cylinder-like shapes in LiDAR, similar to poles/trash bins.
> (2) Learning to reconstruct such geometrically simple shapes provides less discriminative information.
>
> It can also be found that this limitation diminishes with more training data (performance recovers at 100\% setting).
>
> While no links or pdf is allowed during rebuttal, in the revision, we will add this analysis and corresponding visualizations: (1) a zoom-in view of original Pedestrian point clouds and the rendered point clouds from TREND. (2) detection result visualization in low-data regimes (Table 1, 5\% setting).
>
> We will add the discussions and the experiments here to the revision. Thank you again for your suggestions to improve our manuscript!

---

> > ### Author Response · Authors · 2025-08-07
> >
> > Dear Reviewer 4GJv,
> >
> > We hope our response addresses your concerns. We would appreciate it if you can let us know whether there is any pending concern. We are happy for further discussion before the discussion deadline. Thank you.

---

> > ### Comment · Reviewer_4GJv · 2025-08-07
> >
> > Thank the authors for the detailed rebuttal. It addressed my concerns regarding the novelty and the writing quality. I'll maintain my positive rating.

---

> > > ### Author Response · Authors · 2025-08-08
> > >
> > > Dear Reviewer 4GJv,
> > >
> > > Thank you for the reply and appreciation on our work and response.

---

### Official Review · Reviewer_i3pM · 2025-07-03

**Clarity:** 4
**Significance:** 4
**Originality:** 4
**Rating:** 6
**Confidence:** 5

**Summary:**

This paper proposes a novel unsupervised learning method for LiDAR point clouds. The authors note that existing unsupervised representation methods based on auto-encoders or contrastive learning usually require handcrafted set of transformations. In contrast, the proposed approach uses the natural temporal dynamics of the scene as its supervisory signal. By training the model to predict future LiDAR point cloud sequences from past observations, the method implicitly encourages the network to learn object motion, scene dynamics, and semantic information capabilities that are highly valuable for downstream perception tasks. Extensive experiments on multiple standard datasets show that pre-training with TREND yields significant performance gains on downstream 3D object detection and semantic segmentation, outperforming existing SOTA methods.

**Questions:**

1. The paper's motivation highlights that prior unsupervised representation methods implicitly rely on a predefined set of nuisance variability. What are the limitations of these methods and how does the proposed forecasting-based method more effectively and systematically address these shortcomings?

2. In the recurrent embedding scheme described in Section 3.2, the model broadcasts the sinusoidal encoding of ego-motion over the entire feature map and applies a single 3D convolution to evolve temporal features. This design assumes that scene dynamics can be approximated by a single global, rigid transformation. Might this explicit global motion encoding become a bottleneck for the model’s generalization when handling real-world, complex scenes? Could you provide failure-case analyses to show whether the model’s predictions exhibit significant bias when this simplifying assumption breaks down?

**Ethical Concerns:**

["NO or VERY MINOR ethics concerns only"]

**Final Justification:**

1. The author has further clarified the core idea of the paper, explaining that the model combines global information with local features for local evolution via convolutions.
2. The author has provided details on memory consumption and runtime, demonstrating that the proposed method does not introduce significant additional resource overhead.
3. The idea presented in this paper represents a novel and important paradigm in unsupervised point cloud representation, which is highly inspiring for the field.

**Limitations:**

Yes.

**Paper Formatting Concerns:**

No.

**Quality:**

4

**Strengths And Weaknesses:**

Strengths

1. This paper proposes a novel paradigm for LiDAR representation learning, using temporal forecasting as the pretext task for unsupervised pre-training. Unlike common approaches in prior work such as static scene reconstruction or contrastive learning, this method instead utilizes the intrinsic dynamics of the scene as a more natural and information-rich supervisory signal. This could potentially become a highly influential paradigm for unsupervised point cloud learning.

2. The authors point out that both masked auto-encoding and contrastive learning rely on a predefined set of nuisance variability. In contrast, TREND allows the data itself to determine the invariances that need to be learned by simply observing and predicting the scene, an approach that is more fundamental and aligned with the real world. To achieve this, the authors clearly analyze the two major challenges involved and have designed two key technical components in a targeted manner.

3. The authors conducted a comprehensive comparison against a series of strong baseline methods on four mainstream autonomous driving datasets, targeting two different downstream tasks. The experimental results powerfully demonstrate the effectiveness of TREND, especially the significant performance improvements achieved in few-shot learning scenarios, which fully reflects the value and potential of the method.

In summary, this is an outstanding work characterized by its novel concept, rigorous methodology, and solid experimental validation.

Weaknesses

Compared with earlier contrastive learning or auto-encoder frameworks that are structurally simple and computationally efficient, this work additionally introduces the NeRF 3D rendering module. Although this enriches the representation capability, it may increase computational and memory costs during training and inference.

---

> ### Author Rebuttal · Authors · 2025-07-29
>
> Dear Reviewer i3pM,
>
> Thank you for your precious time on the review and your constructive suggestions to improve our manuscript! We appreciate the acknowledgment that TREND is a novel unsupervised 3D representation learning method, the experiments are extensive , the results are strong, the research direction is fundamental and promising and writing is clear.
>
> We provide further discussions on your questions as belows:
>
> **Q1: TREND may increase computational and memory costs during training and inference.**
>
> Thank you for this important concern. We provide detailed analysis divided into pre-training and fine-tuning phases:
>
> ***Pre-training Costs:*** While TREND introduces temporal forecasting and neural field rendering, the actual memory costs are comparable to baseline methods. All pre-training methods utilize approximately the same GPU memory as we utilize several design choices when sampling the rendering rays (Line 203-206 in the paper):
> (1) Ground point filtering: We remove less informative ground points using sensor height as thresholds, which is originally provided in the datasets.
> (2) Uniform ray sampling: After filtering, we uniformly sample $N_{render}$=12,288 rays per frame rather than processing all points.
>
> As for computational cost during pre-training, since TREND employs Recurrent Embedding scheme, TREND requires approximately 8% more time than previous methods per epoch (65 mins v.s. 60 mins on 8-A100) for pre-training, which is feasible.
>
> ***Fine-tuning (Inference) Costs:*** Recurrent embedding and temporal neural fields are only needed during pre-training stage. The downstream model architecture, computational cost, and memory usage are identical across all methods during both fine-tuning and inference. When we are fine-tuning for downstream perception tasks, we load the pre-trained 3D encoder weights to the same model architecture and train the same detector/segmentation model.
>
> This design ensures that while we leverage rich temporal information during pre-training, practical deployment remains efficient and scalable.
>
> **Q2: The paper's motivation highlights that prior unsupervised representation methods implicitly rely on a predefined set of nuisance variability. What are the limitations of these methods and how does the proposed forecasting-based method more effectively and systematically address these shortcomings?**
>
> Thank you for this insightful question. Nuisance variability refers to variables inherent in the input that should be non-consequential to the outcome, but nonetheless may impact the output. An example of this is orientation: the same object appearing in different orientations can cause the outcome to differ. To obtain the same outcome, one needs to be invariant. The most common way is by choosing the set of augmentations. The key limitation of existing methods is their reliance on manually designed augmentations (pre-defined nuisance variable set).
>
> ***Limitations of existing methods with pre-defined nuisance variable set.***
> (1) Contrastive learning: Manually selects augmentations (rotations, translations) as positive pairs, potentially missing important invariances or including irrelevant ones.
> (2) Masked auto-encoding: Assumes occlusion is the primary nuisance variable, ignoring other dynamic factors.
>
> The pre-defined nuisance variable set is what they assume important for downstream tasks. This makes the learned representation limited to specific assumption.
>
> ***TREND's advantages.*** Instead of pre-defining invariances (specific assumption), we let natural scene dynamics determine what should be the set of invariant. Through temporal forecasting, occlusion, orientation, and translation invariances emerge naturally from observing object motion. Also, the model learns that points belonging to the same object instance move coherently, encoding semantic relationships. And TREND also learns dynamic interactions between ego-vehicle and other traffic participants.
>
> This data-driven approach to invariance learning is more robust and generalizable than handcrafted transformations, as evidenced by our consistent improvements across diverse datasets and tasks.
>
> **Q3: In the recurrent embedding scheme described in Section 3.2, the model broadcasts the sinusoidal encoding of ego-motion over the entire feature map and applies a single 3D convolution to evolve temporal features. This design assumes that scene dynamics can be approximated by a single global, rigid transformation. Might this explicit global motion encoding become a bottleneck for the model’s generalization when handling real-world, complex scenes? Could you provide failure-case analyses to show whether the model’s predictions exhibit significant bias when this simplifying assumption breaks down?**
>
> Thank you for this detailed question. We want to clarify the design and provide additional analysis:
>
> ***Clarification of the approach***: The recurrent embedding does not assume a single global rigid transformation. In the recurrent embedding scheme, we first use Sinusoidal Encoding to embed ego-vehicle motion ($\Delta x$, $\Delta y$, $\Delta \theta$). Then we broadcast the ego-vehicle motion feature to each spatial location and concatenate it with local features. Next, a shallow 3D convolution (3 layers) is applied to predict the future features over the entire 3D scene conditioned on the ego-vehicle motion.
>
> While local features reflect the understanding of other traffic participants and the environment, this concatenation provides local features with understanding of ego-vehicle motion. Despite the feature vector containing the vehicle ego-motion, the remainder of the feature vector allows us to predict the feature evolution. The recurrent embedding allows us to model the evolution of the latent scene features based on vehicle ego-motion. Thus, the mechanism behind it is not a single warping and it does not assumes that scene dynamics can be approximated by a single global, rigid transformation.
>
> ***Empirical validation***: Our consistent improvements across datasets (Once, NuScenes, SemanticKITTI) with diverse scene complexities demonstrate robustness. The method generalizes well across different geographical locations and various traffic scenarios.
>
> Nonetheless, our approach assumes that the data is representative of realistic scenarios. Hence, when it comes to transferring representation learned in Once dataset to downstream task in Waymo dataset (Table 3 in the paper), it can be found that TREND achieves less performance improvement compared to downstream task in Once dataset. However, our approach still improves over existing methods.
>
> We will add the discussions here to the revision. Thank you again for your suggestions to improve our manuscript!

---

> > ### Comment · Reviewer_i3pM · 2025-08-05
> >
> > The author has answered my questions very comprehensively, especially regarding my concerns about computational cost. The response clarifies that the proposed method does not introduce additional memory overhead, and the computation time is acceptable. The author also cleared up my misunderstanding of the global rigid transformation, explaining that the paper combines global information with local features and uses convolutions for local evolution. Therefore, I believe this is an outstanding paper in terms of both the novelty of its idea and the completeness of its experiments.
> >
> > It is an excellent work in the field of unsupervised point cloud representation learning, and I am willing to raise my rating for the paper.

---

> > > ### Author Response · Authors · 2025-08-05
> > >
> > > Dear Reviewer i3pM,
> > >
> > > Thank you for the comments, reply and appreciation on our work and response. Thank you again for your constructive suggestions to improve the paper.

---

### Note · Authors · 2025-08-12

We sincerely appreciate the efforts of all reviewers, ACs, SACs, PCs to review, provide constructive suggestions and discussions to improve our paper. We are grateful that all reviewers acknowledge that our response resolve all their concerns about the paper. We will incorporate the discussion and new experiments to the revision to further improve the quality of our paper.

We also appreciate that reviewers acknowledge TREND is a novel unsupervised 3D representation learning method (Reviewer i3pM, g95E, jBSj) (Reviewer 4GJv after rebuttal), the experiments are extensive (Reviewer i3pM, 4GJv, g95E, jBSj), the results show the effectiveness of TREND (Reviewer i3pM, 4GJv, g95E, jBSj), the research direction is fundamental and promising (Reviewer i3pM, g95E) and well-motivated (Reviewer 4GJv), writing is clear and well-structured (Reviewer i3pM, g95E).

---

### Decision · Program_Chairs · 2025-09-17

**Decision:**

Accept (spotlight)

**Comment:**

This paper proposes a novel self-supervised 3D representation learning by forecasting future observations from temporal LiDAR data. Unlike previous methods, the proposed approach leverages natural temporal dynamics of the scene as its supervisory signal and enables to learn effective representations from scene dynamics, which are effective for downstream perception tasks. This paper received positive ratings from all reviewers; all reviewers agree that the paper tackles an important problem, proposes an interesting approach, and shows significant improvement over previous methods. The rebuttal also addresses most of the reviewer’s concerns, in particular, the computational cost of the proposed method. The AC also appreciates the scientific contribution of the novel unsupervised learning method for LiDAR point clouds, and thus recommends accepting this work.